



# Solar FTIR measurements of NO$_x$ vertical distributions: Part II) Experiment-based scaling factors describing the diurnal increase of stratospheric NO$_2$ and NO

Pinchas Nürnberg[1], Sarah A. Strode[2,3], and Ralf Sussmann[1]

[1]Karlsruhe Institute of Technology, IMK-IFU, Garmisch-Partenkirchen, Germany
[2]Goddard Earth Sciences Technology and Research (GESTAR-II), Morgan State University, Baltimore, MD, 21251 USA
[3]NASA Goddard Space Flight Center, Greenbelt, MD 20771, USA

*Correspondence to*: Pinchas Nürnberg (pinchas.nuernberg@kit.edu)



**Abstract**

Long-term experimental stratospheric $NO_2$ and NO partial columns measured by means of solar Fourier-transform infrared (FTIR) spectromertry at Zugspitze (47.42° N, 10.98° E, 2964 m a.s.l.), Germany were used to create a set of experiment-based monthly scaling factors ($SF_{exp}$). The underlying data set is published in a companion paper (Nürnberg et al., 2023) comprising over 25 years of measurements depicting the diurnal variability of stratospheric $NO_2$ and NO partial columns in dependence

of local solar time (LST). In analogy to recently published simulation-based scaling factors by Strode et al. (2022), we created $SF_{exp}$ normalized to local solar noon for $NO_2$ and NO for every month of the year as a function of solar zenith angle (SZA). Beside a boundary value problem at minimum SZA values originating in averaging over different times of the month, the obtained scaling factors $SF_{exp}(NO_2)$ and $SF_{exp}(NO)$ in dependence of SZA represent very well the diurnal behavior already shown in model simulations and experiment in the literature. This behavior is a well pronounced increase of the $NO_2$ and NO

stratospheric partial colum with the time of the day and a flattening of this increase after noon. In addition to the discussion of $SF_{exp}$, we validate the simulation-based scaling factors $SF_{sim}(NO_2)$ (Strode et al., 2022) and present simulation-based scaling factors for NO $SF_{sim}(NO)$. The simulation-based scaling factors show an excellent agreement with our the experiment-based ones, i.e. for $NO_2$ and NO the mean bias of the modulus between experiment and simulation over all SZA and months is only 0.02 %. We show, that recently used model simulations can describe very well the real behavior of nitrogen oxide ($NO_x$)

variability in the stratosphere. Furthermore, we conclude that ground-based FTIR measurements can be used for validation of the output of photochemistry models as well as creating experiment-based data sets describing the diurnal stratospheric $NO_x$ variability in dependence of SZA. This is a contribution to improved satellite validation and a better understanding of stratospheric photochemistry.



## 1 Introduction

The important role of $NO_2$ and $NO$ in stratospheric photochemistry has been known for half a century (Crutzen, 1979). Both nitrogen oxides ($NO_x$) are a product of the photolysis of $N_2O$ and are an important part of the ozone ($O_3$)-destroying nitrogen catalytic cycle which controls the $O_3$ abundance in the stratosphere (Johnston, 1992). Additionally, $NO_x$ is a product of industry and traffic in the troposphere. Especially in urban areas, it can serve as a precursor for e.g. $O_3$ or nitric acid ($HNO_3$) and

35 therefore promote smog events and directly affect human health (World Health Organization. Regional Office for Europe, 2003). Furthermore, $NO_2$ has the potential to cause significant radiative forcing during pollution events with highly elevated $NO_2$ concentrations in the troposphere (Solomon et al., 1999).

The monitoring and quantification of $NO_x$ total columns has been conducted since 1967 via different satellite missions (Godin-Beekmann, 2010; Rusch, 1973). For the observation of tropospheric pollution events (e.g. smog), therefore, the knowledge of

40 the stratospheric contribution to the total column is crucial. One way to face this problem is the reference sector method, taking unpolluted total columns at a similar latitude (e.g. above the ocean) as a reference and subtract it from the total column (Richter and Burrows, 2002). The two main assumptions justifying this approach are the longitudinal homogeneity of the stratospheric column and negligible tropospheric columns over the ocean. However, due to the strong diurnal cycle of $NO_2$ and $NO$ no time mismatch should occur between both columns.

One method to face the problem of time and site mismatches when comparing different $NO_x$ columns is the use of ground-based Fourier-transform infrared (FTIR) measurements. This method can provide data from any time of the day during sun light hours, giving the opportunity to describe diurnal $NO_x$ variabilities with a high precision as done for $NO_2$ by Sussmann et al. (2005). For the first time, they found a reliable diurnal $NO_2$ increasing rate of $(1.02 \pm 0.12) \cdot 10^{14}$ $cm^{-2}$ $h^{-1}$ derived from FTIR measurements at mid-latitudes. Additionally, the retrieved FTIR data can have a certain altitude resolution, which allows

conclusions about $NO_x$ partial column variabilities, e.g. of the stratospheric columns (Zhou et al., 2021; Yin et al., 2019). In Part 1 of our two companion papers (Nürnberg et al., 2023) we used these advantages of ground-based FTIR measurements to retrieve stratospheric partial columns from long-term $NO_2$ and $NO$ measurements above Zugspitze (47.42° N, 10.98° E, 2964 m a.s.l.), Germany, yielding information on $NO_x$ diurnal variability for every month of the year. This specific data set has the potential to improve satellite validation and can serve as a basis for the description of stratospheric $NO_x$ variabilities

with high time resolution. However, the data from ground-based measurements can only be received for the limited number and locations of existing sites.

A method without this site restriction describing stratospheric $NO_x$ concentrations with global coverage is the use of model data from three-dimensional global transport and photochemistry models. The latter are able to describe trace gas concentrations in dependence of altitude, latitude and longitude with a very good time resolution. In comparison to one-

60 dimensional models describing only the vertical distribution of atmospheric trace gases (e.g. $O_3$, $NO_2$, $NO$) (Allen et al., 1984; Prather and Jaffe, 1990), three-dimensional models simulate transport fluxes in all three dimensions and are able to include nearly all feedback mechanisms of the real world (Mclinden et al., 2000; Chang and Duewer, 1979). Both types of models can account for diurnal variabilities and have been used in the last decades for inter-satellite comparisons (Brohede et al., 2007; Dubé et al., 2020) as well as for satellite data validation (Bracher et al., 2005) and correction (Dubé et al., 2021; Wang et al.,

2020). However, these studies differ from case to case and do not provide general global information about $NO_x$ variability. These global information should be site independent and can be applied to any satellite validation or correction all over the planet.

Here, a recent study of Strode et al. (2022) closed this lack by developing a set of simulation-based scaling factors ($SF_{sim}$), which describe the diurnal variability of $NO_2$. A given $SF_{sim}$ is a measure for the change of trace gas concentrations during the

70 day normed to a specific time (here sunrise or sunset). $SF_{sim}$ are extracted from a three-dimensional model, which considers long-range transport, stratospheric and tropospheric chemistry as well as aerosol radiation and transport. The generated monthly output is available for latitudes between -90° and 90° (1° steps) and altitudes between 6 km and 78 km (0.5 km steps)



for every time of the day given in solar zenith angle (SZA) values (Strode et al., 2022). This extensive research opens up the opportunity for the comparison, validation, and correction of remote and ground-based data products, by overcoming time or

site mismatches.

However, an observational counterpart, i.e. an analogous data set of experiment-based scaling factors describing the diurnal increase of stratospheric $NO_x$ still does not exist, due to the lack of reliable long-term data comprising the full diurnal $NO_2$ and NO variability. To close this lack, in this paper we create a set of experiment-based scaling factors ($SF_{exp}$) in analogy to the simulation-based scaling factors published by Strode et al. (2002). On the one hand, this data set should serve as a general set

of data describing the $NO_x$ diurnal variability in dependence of SZA for the given latitude (47° N) of our observation site. On the other hand, we would like to use it to validate the recently published model data for $SF_{sim}(NO_2)$ (Strode et al., 2022) as well as validate unpublished model data for $SF_{sim}(NO)$ (Sarah Strode, personal communication, 2023). For this $SF_{exp}$ data set we will use the observational results described in Part 1 of our set of two companion papers (Nürnberg et al., 2023), where a reliable long-term data set of $NO_2$ and NO partial columns above 16 km altitude above Zugspitze was created. As described

above, these long-term data are retrieved from ground-based FTIR measurements and describe the diurnal variability of stratospheric $NO_x$ within timesteps of minutes for every month of the year.

This paper (as Part 2 of our two companion papers) briefly describes in Sect. 2 the experimental set up and the resulting FTIR data taken from Part 1 (Nürnberg et al., 2023). In Sect. 3, the dependence on SZA for $NO_2$ and NO is shown and the resulting diurnal variations presented in detail in Part 1 are discussed shortly, before the $NO_x$ partial columns (> 16 km) are converted

into experiment-based scaling factors ($SF_{exp}(NO_2)$ and $SF_{exp}(NO)$) in Sect. 4. Finally, the resulting $SF_{exp}$ are compared qualitatively and quantitatively to $SF_{sim}$ retrieved from model simulations.

## 2 Used FTIR data

All data of this study are retrieved from long-term ground-based FTIR solar absorption measurements at the Zugspitze, Germany (47.42° N, 10.98° E, 2964 m a.s.l.). The high-altitude observatory at Zugspitze is located in the German alps and can

be considered as a clean site without strong influences from pollution events in the boundary layer. The used Bruker IFS 125HR spectrometer is operated continuously since 1995 at the Zugspitze. The experimental set-up and retrieval strategy are described in our part I) companion paper (Nürnberg et al., 2023). The pollution filtered NO and $NO_2$ stratospheric partial columns (above 16 km altitude) derived in our part I) study serve as a basis for the experiment-based scaling factors created now in this part II) work. The data set comprises 6,213 NO and 16,023 $NO_2$ partial columns measured at the Zugspitze between

1995 and 2022.

## 3 Experimental data

### 3.1 $NO_x$ stratospheric partial column dependence on SZA

Figure 1 shows the $NO_2$ stratospheric partial columns (black symbols) taken from Nürnberg et al. (2023) for every month as a function of SZA. Note this is the same data as shown in our Part 1 (Fig. 3 therein), which had been therein plotted as a function

of local solar time. The $x$-axis is interrupted for SZA values not existing in the respective month. Here, we define SZA to be positive in the morning from sunrise (SZA = 90°) to local solar noon (respective minimum value dependent of the season) and to be negative in the afternoon between local solar noon and sunset (SZA = -90°).

As already described and discussed in Part 1 of the two companion papers, the diurnal increase of the $NO_2$ stratospheric partial column follows for every month a linear behavior from sunrise to sunset. Briefly, this behavior reflects the photolysis of the

reservoir species $HNO_3$ and $N_2O_5$ resulting in a consecutive increase of $NO_2$ during daytime (Crutzen, 1970).



Figure 2 shows in a similar way the NO stratospheric partial columns (black symbols) taken from the same work for every month in dependence of SZA (Nürnberg et al., 2023). Note this is the same data as shown in our Part 1 (Fig. 5 therein) as a function of local solar time. Briefly, the data show the typical diurnal increase of stratospheric NO described in the literature via model calculations (Dubé et al., 2020; Mclinden et al., 2000) or shown experimentally (Zhou et al., 2021; Rinsland et al., 1984) for every month. Here, the photolysis of the reservoir species $N_2O$ leads to a well-pronounced increase of stratospheric NO concentration in the morning (Crutzen, 1970). After local solar noon, the shift of the $NO_2$-NO equilibrium, the increasing amount of $O_3$ and the solar elevation dependency of the involved photochemical reaction lead to a strong flattening of the diurnal NO curve in dependence of SZA in comparison to $NO_2$. This afternoon-effect is more pronounced in the summertime (mid row) than the rest of the year (Nürnberg et al., 2023).

## 4 Calculation of experiment-based scaling factors

A set of experiment-based scaling factors ($SF_{exp}$) in analogy to the model-based scaling factors ($SF_{sim}$) published by Strode et al. (2022) was created as follows: The mean values for 2° bins of SZA of the stratospheric partial column (> 16 km) were calculated. In a next step, these mean values were normalized to the minimum SZA at month $15^{th}$ resulting in monthly $SF_{exp}$ sets for $NO_2$ and NO shown in Fig. 3 and Fig. 4, respectively. The (differing) SZAs used for normalization for the individual months can be found in the respective legends. They are the minimum SZA at day 15 of the respective month. These data reflect the diurnal variation of stratospheric $NO_2$ and NO above Zugspitze, Germany. Values resulting from only one measurement point are shown in red without error bar.

$SF_{exp}(NO_2)$ (Figure 3, black and orange symbols) follows every month a linear diurnal trend, reflecting the increase in stratospheric $NO_2$ concentration. There are two observations which can be pointed out here. First, the error bars in Fig. 3 (i.e. ±2 standard errors of the mean, ±2 SEM = ±2 $\sigma/\sqrt{(n)}$) are independent of the season and are very small, reflecting a low scattering within the 2° SZA bins and enough averaging data points $n$. Second, in spring and autumn, at local solar noon (minimum SZA), a significant increase in $SF_{exp}(NO_2)$ is visible. This effect can be understood as a boundary value problem being due to the relatively fast change of SZA and of the $NO_2$ stratospheric partial column (seasonal variation) during the spring and autumn months, respectively. Here, the combination of both, the SZA and stratospheric partial column changes within one month end up with an increased averaged $NO_2$ stratospheric partial column near the minimum SZA. The reason is that for SZA values below the minimum SZA of each month $15^{th}$, only partial columns from one half of the month can contribute to the average. Unfortunately, the stratospheric partial columns of this half deviate significantly from the monthly mean. Figure S1 in the supporting material illustrates this phenomenon using the $NO_2$ partial column above 16 km altitude. Here, the first half (red symbols) and the second half (blue symbols) of April is split up into two datasets underlining the described boundary layer problem. At low SZA values, only blue data points sum up to the averaged values, considering only the second half of the month. Consequently, the partial column and of course the scaling factor increases artificially. This effect leads us to the exclusion of these data points (Figure 3, orange symbols) below the minimum SZA reached at day 15 of the respective month. The whole used data set of $SF_{exp}(NO_2)$ can be found in the supporting material Table S1-S4.

For $SF_{exp}(NO)$ (Figure 4, black and orange symbols), the difference in diurnal increase in comparison to $NO_2$ is very well pronounced. Before local solar noon, $SF_{exp}$ increases for every month linearly. After local solar noon, the described flattening of the increase is visible. Here, the NO stratospheric partial column stays almost constant within the scattering until sunset independent of the season. The ±2 SEM error bars of $SF_{exp}(NO)$ shown in Fig. 4 are also very small, but more values are excluded (red symbols) due to the availability of only one measurement point within the corresponding 2° SZA bin. This reflects the lower data base of the NO retrieval, originated in the use of another spectral micro-window for analysis. However, the small error bars underline, that for most of the mean values, the data base is reliable. Near local solar noon for $SF_{exp}(NO)$ a similar but even less pronounced effect can be seen as described for $NO_2$ before. Here, the deviation from the visible trend





at spring or autumn months is very small. However, for consistent data handling we will also exclude the respective values (orange symbols) for $SF_{exp}(NO)$ below the minimum SZA at each month $15^{th}$. The whole used data set of $SF_{exp}(NO)$ can be found in the supporting material Table S5-S8.

### 5 Model comparison of NO$_x$ scaling factors

In the previous section, we created experiment-based averaged monthly scaling factors $SF_{exp}$ for $NO_2$ and $NO$ describing the diurnal variation of stratospheric $NO_x$ concentration above Zugspitze, Germany. Next, we will compare the discussed results for $SF_{exp}$ to model-based scaling factors $SF_{sim}$ for $NO_2$ published by Strode et al. (2022) and for $NO$ calculated from the same GEOS-GMI model simulation as the $NO_2$ scaling factors. Details of the GEOS model simulation with GMI chemistry (Duncan et al., 2007; Strahan et al., 2007; Nielsen et al., 2017) are described in Strode et al. (2022) and refs therein.. The model parameters and the analysis method can be found in the literature (Strode et al., 2022). The given scaling factors $SF_{sim}(NO_2)$ and $SF_{sim}(NO)$ are available for 146 levels between 6 km and 78.5 km altitude in a 0.5 km grid and are normed to SZA = 90° (sunrise). For a better comparison of experiment and model, we calculated mean values for $SF_{sim}$ which also represent the stratospheric partial column above 16 km altitude. In order to do so, for each model level $z$, $SF_{sim}(z)$ was weighted to the mean monthly partial column profile of the given $NO_x$ retrieval at $z$ and $SF_{sim}(> 16 \text{ km})$ was obtained via averaging over $SF_{sim}(16 \text{ km})$ to $SF_{sim}(78.5 \text{ km})$. Furthermore, $SF_{sim}(> 16 \text{ km})$ was also normalized to the minimum SZA (rather than sunrise/sunset) at month $15^{th}$ as done for $SF_{exp}$ in Sect. 4.

$SF_{sim}(NO_2)$ and $SF_{sim}(NO)$ are additionally shown in Fig. 5 and Fig. 6, respectively (red line). At first appearance, $SF_{exp}$ (black symbols) and $SF_{sim}$ (red line) fits together very well and the model data follow the experimental diurnal variation for both species $NO_2$ and $NO$.

#### 5.1.1 Quantitative evaluation

For the quantitative evaluation of the model comparison, the residuals between experiment and model $(SF_{exp}-SF_{sim})/SF_{sim}$ are calculated for $SF(NO_2)$ and $SF(NO)$ and are shown in Fig. 7 and Fig. 8, respectively. Additionally, the mean bias per month is shown as a mean value over all SZA (red dotted line).

The residuals of $SF(NO_2)$ (Figure 7) show over the whole season a very good agreement between experiment and model within $\pm 0.2$ %, reflecting the high quality of the GEOS GMI simulation at midlatitudes. Only for a few months, significant differences between experiment and model are visible at high SZA values (near sunrise). For August, September and October, the morning increase of $NO_2$ is less pronounced in the model, leading to a significant deviation from the experimental values and an overestimation of the experiment-based scaling factors $SF_{exp}$. However, the experimental values describing the stratospheric $NO_2$ variability can be also influenced by tropospheric variations, because the used $NO_2$ partial column cannot be treated as completely independent of the tropospheric partial column (see Nürnberg et al. (2023)).

Table shows the mean bias (see also Figure 7, red dotted line) for every month calculated from the residuals shown in Fig. 7 together with two times the SEM $(2 \sigma/\sqrt(n))$. Unfortunately, due to the small values of 2 SEM of 0.0063 % to 0.0193 % for most of the months (except January, February, Jun and December), 2 SEM is smaller than the mean bias. Therefore, when taking 2 SEM as a quantitative indicator, $SF_{exp}$ and $SF_{sim}$ agrees only in four months within the margin of error. However, when considering the mean deviation between experiment and model of below $|0.0444 \%|$ per month, we can state that the model data published by Strode et al. (2022) reflect sufficiently well the experimental values retrieved from solar FTIR measurements at midlatitudes.

A very similar behavior can be obtained for $SF(NO)$ (Figure 8). With a maximum deviation of $\pm 0.2$ % the agreement between experiment and model is very similar as seen for $NO_2$. However, it is remarkable, that for specific months (January, February, August, September, October, December) the last data points nearest to sunrise (high SZA region) deviate significantly from





zero. Comparing to Fig. 6, the experimental values in this region seems not to follow the continuous decrease expected from model descriptions. The NO increase in the morning is more pronounced in the model, leading to a significant deviation from the experimental values and an underestimation of the experiment-based scaling factors $SF_{exp}$. In the same manner as discussed

before for $NO_2$, the experimental values describing the stratospheric NO variability can be influenced by tropospheric variations, because the used NO partial column cannot be treated as completely independent of the tropospheric partial column (see Nürnberg et al. (2023)).

In the same way as done for $NO_2$, the mean bias (see also Fig. 8, red dotted line) and $2\,\sigma/\sqrt{(n)}$ (2 SEM) are calculated and are shown in Table for the NO residuals. Here, a better agreement between experiment and model can be quantified. For seven

200     months (January, February, March, April, May, November, December) the mean bias is smaller than 2 SEM indicating an agreement between experiment and model within the error bars. Nevertheless, this observation not only reflects a better agreement between experiment and model but can be also explained with a higher scattering of the residuals leading to a higher SEM. This can be confirmed when comparing the values for 2 SEM given in Table and Table. With a mean 2 SEM of the residuals over all months of 0.0093 % for $NO_2$ and 0.0191 % for NO, respectively, the residual scattering with a similar $n$ and

a similar mean bias of 0.02 % is two times larger for NO.

In conclusion, the quantitative comparison of the experimental derived scaling factors $SF_{exp}$ and the scaling factors derived from model simulations $SF_{sim}$ for $NO_2$ and NO showed very good agreement of both data sets with a mean bias between experiment and model of only 0.02 % over all months underlining the quality of the model data at midlatitudes and the reliability of the retrieved experiment-based scaling factors.



**6 Summary and Conclusions**

In this work, we reanalyzed an experimental long-term data set from solar FTIR measurements over 25 years of measurement at the Zugspitze (47.42° N, 10.98° E, 2964 m a.s.l.), Germany, published along in a companion paper (Part 1, Nürnberg et al., 2023) . We present for the first time experiment-based scaling factors $SF_{exp}$ in dependence of the solar zenith angle (SZA) representing monthly diurnal $NO_2$ and NO variabilities in the stratosphere (> 16 km altitude) within timesteps of minutes. $SF_{exp}$ is a measure for the variability of the $NO_x$ partial column above 16 km altitude in comparison to local solar noon. We calculated $SF_{exp}$ from the time dependent monthly $NO_x$ partial columns (published in Part 1) by averaging over SZA bins of 2° and a normalization to the minimum SZA at day 15 of the respective month. The resulting values of $SF_{exp}(NO_2)$ and $SF_{exp}(NO)$ reflect very well the expected diurnal variability of $NO_2$ and NO described in Part 1 (Nürnberg et al., 2023). Only the boundary values in spring and autumn months deviate significantly due to the relatively fast change of the minimum SZA during these months influencing the average value. Neglecting these values leads to two reliable experiment-based data sets for $SF_{exp}(NO_2)$ and $SF_{exp}(NO)$. Furthermore, we used these new experiment-based data sets to validate recently published simulation-based scaling factors $SF_{sim}(NO_2)$ (Strode et al., 2022) and recently simulation-based scaling factors $SF_{sim}(NO)$ from a global study representing a similar latitude (47 °N). Comparing experiment and model simulation, we find an excellent agreement for stratospheric $NO_2$ and NO diurnal variabilities with a mean bias of the modulus over all months and SZA of only 0.02 % with no significant deviating trends for boundary values. These results underline the quality of recent multi-dimensional model simulations of stratospheric trace gases, representing very well experimental data. Additionally, we showed, that ground-based FTIR measurements can provide reliable information about stratospheric $NO_x$ variability within time steps of minutes, which can serve as a good basis for the validation of global model simulations and therefore can help to further optimize satellite validations.

The analysis method of the retrieval of stratospheric $NO_2$ and NO partial columns over Zugspitze, Germany, published in Part 1 of the two companion papers (Nürnberg et al., 2023) in combination with the generalization of this data by calculating unitless scaling factors $SF$ and the validation of recently published model data in this paper (Part 2) can be seen as a strong tool for the further validation and correction of global model and satellite data. This approach can be taken for any ground-based FTIR spectrometer generating a global set of experiment-based stratospheric $NO_2$ and NO partial columns or scaling factors $SF_{exp}(NO_2)$ and $SF_{exp}(NO)$.

**Data availability**

The presented calculated experimental factors $SF_{exp}$ can be found in the supporting material of this paper. The used experimental data is published along in Part 1 of the two companion papers (Nürnberg et al., 2023). Any other data of interest underlying this publication can be obtained at any time from the corresponding author on demand. The simulated scaling factors for NO2 and NO are available at this website: https://avdc.gsfc.nasa.gov/pub/data/project/GMI_SF/

**Competing Interests**

None.

**Acknowledgements**

Funding by the Federal Ministry of Education and Research of Germany within the Project ACTRIS-D (grant 01LK2001B) is gratefully acknowledged. We acknowledge funding by the Helmholtz Research Program "Changing Earth – Sustaining our Future" within the Research Feld "Earth and Environment" and by the KIT-Publication Fund of the Karlsruhe Institute of



Technology. SAS acknowledges support from NASA grant 80NSSC18K0711, the NASA Modeling, Analysis, and Prediction (MAP) Program, and computing resources from the NASA Center for Climate Simulation (NCCS) for the simulated scaling factors.



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



**Table 1.** Calculated mean bias of residuals ($[SF_{exp}\text{-}SF_{sim}]/SF_{sim}$) for every month between experiment and simulations for $NO_2$ and the standard error of the mean ($\sigma/\sqrt{(n)}$) of this value.

| Month | J (%) | F (%) | M (%) | A (%) | M (%) | J (%) | J (%) | A (%) | S (%) | O (%) | N (%) | D (%) |
|---|---|---|---|---|---|---|---|---|---|---|---|---|
| mean bias | 0.0065 | -0.0050 | -0.0438 | -0.0364 | -0.0071 | 0.0082 | -0.0194 | -0.0404 | -0.0280 | -0.0444 | -0.0407 | -0.0138 |
| $2\sigma/\sqrt{(n)}$ | 0.0136 | 0.0094 | 0.0084 | 0.0076 | 0.0063 | 0.0085 | 0.0076 | 0.0087 | 0.0076 | 0.0069 | 0.0081 | 0.0193 |
| bias < 2SEM? | Yes | Yes | No | No | No | Yes | No | No | No | No | No | Yes |

**Table 2.** Calculated mean bias of residuals ($[SF_{exp}\text{-}SF_{sim}]/SF_{sim}$) for every month between experiment and simulations for NO and 2 times the standard error of the mean ($2\,\sigma/\sqrt{(n)}$) of this value.

| Month | J (%) | F (%) | M (%) | A (%) | M (%) | J (%) | J (%) | A (%) | S (%) | O (%) | N (%) | D (%) |
|---|---|---|---|---|---|---|---|---|---|---|---|---|
| mean bias | 0.0060 | 0.0126 | -0.0105 | -0.0028 | 0.0008 | 0.0164 | 0.0206 | 0.0397 | 0.0556 | 0.0316 | -0.0096 | 0.0150 |
| $2\sigma/\sqrt{(n)}$ | 0.0335 | 0.0254 | 0.0168 | 0.0112 | 0.0107 | 0.0163 | 0.0160 | 0.0115 | 0.0124 | 0.0144 | 0.0179 | 0.0425 |
| bias < 2SEM? | Yes | Yes | Yes | Yes | Yes | No | No | No | No | No | Yes | Yes |

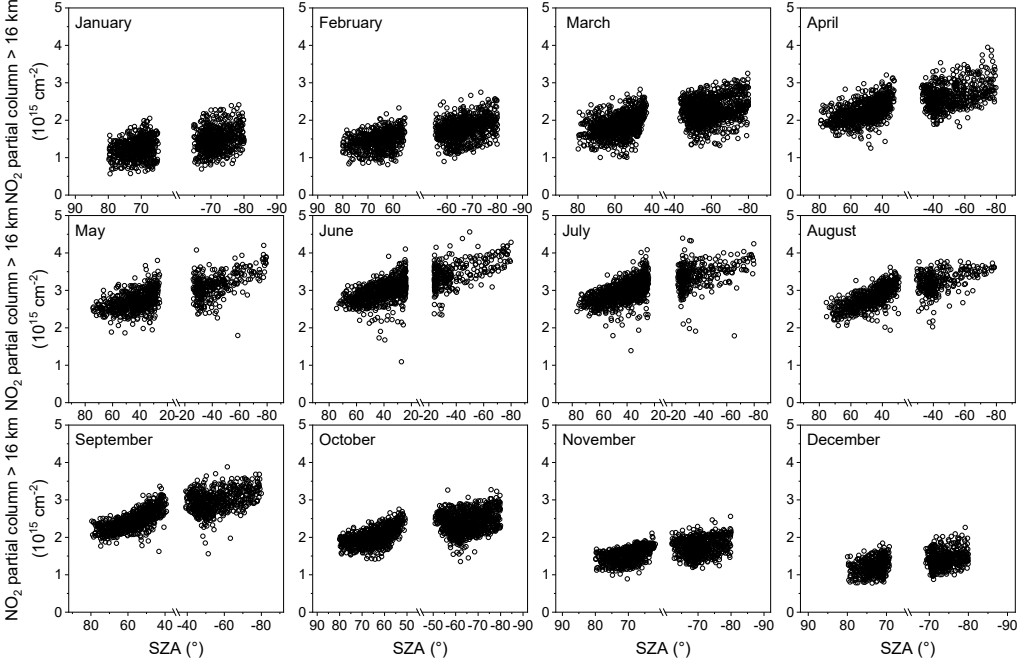

**Figure 1.** Retrieved $NO_2$ partial column above 16 km altitude measured at Zugspitze (black symbols) for every month in dependence of
330 SZA.



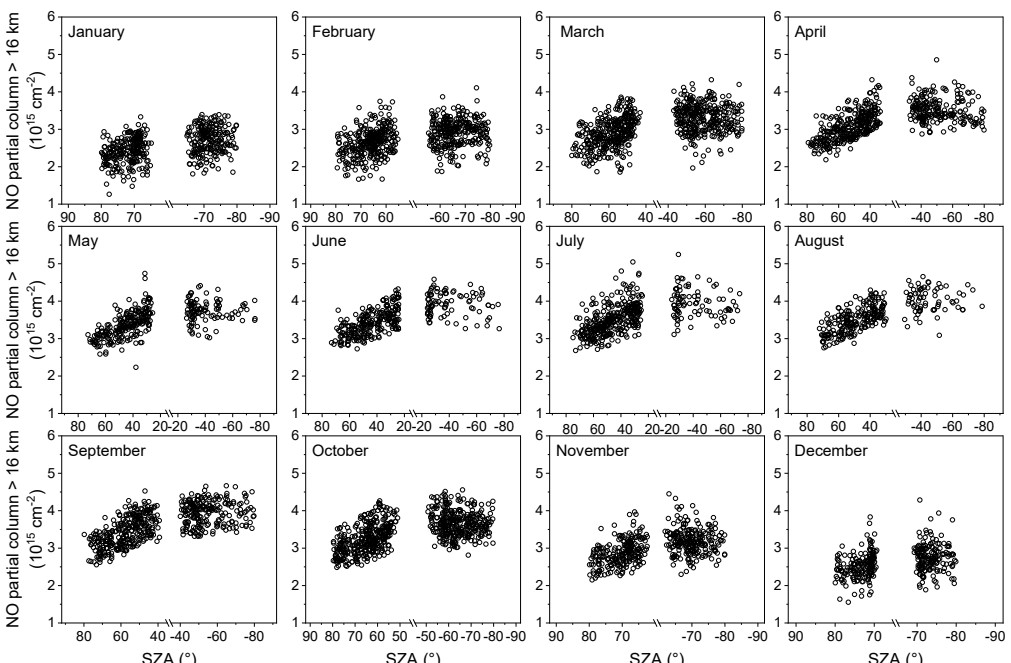

**Figure 2.** Retrieved NO partial column above 16 km altitude measured at Zugspitze (black symbols) for every month in dependence of SZA.

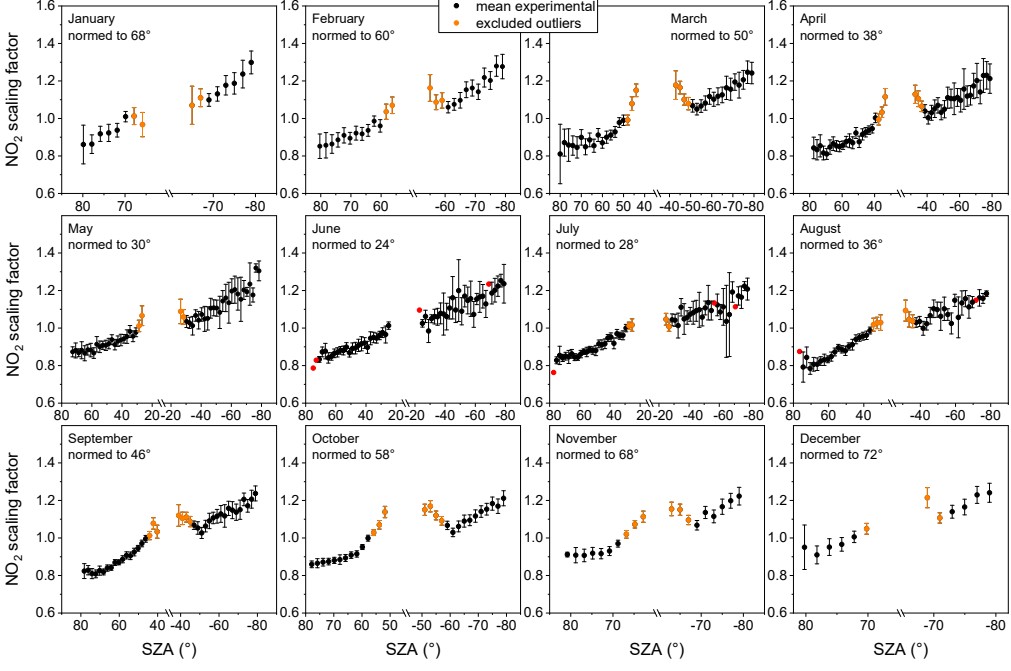

**Figure 3.** Calculated normed $NO_2$ scaling factors $SF_{exp}(NO_2)$ above 16 km altitude measured at Zugspitze (black; orange symbols are excluded outliers) for every month in dependence of the SZA. The values represent the mean value within 2° SZA bins. The error bars represent two times the standard error of the mean ($\pm 2\ \sigma/\sqrt{(n)}$) value. Values resulting from only one measurement point are shown in red without error bar. The SZA used for normalization for the respective month for experiment and model is given in each legend.





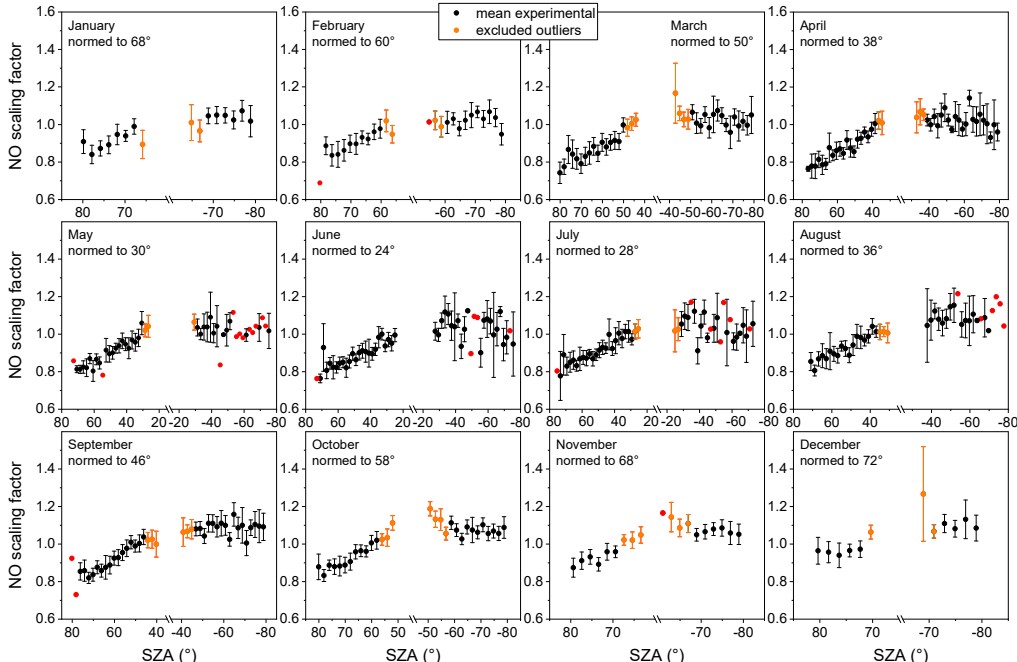

**Figure 4.** Calculated normed NO scaling factors $SF_{exp}$(NO) above 16 km altitude measured at Zugspitze (black; orange are excluded outliers) for every month in dependence of SZA. The values represent the mean value within 2° SZA bins. The error bars represent two times the standard error of the mean ($\pm 2\ \sigma/\surd(n)$) value. Values resulting from only one measurement point are shown in red without error bar. The SZA used for normalization for the respective month for experiment and model is given in each legend.





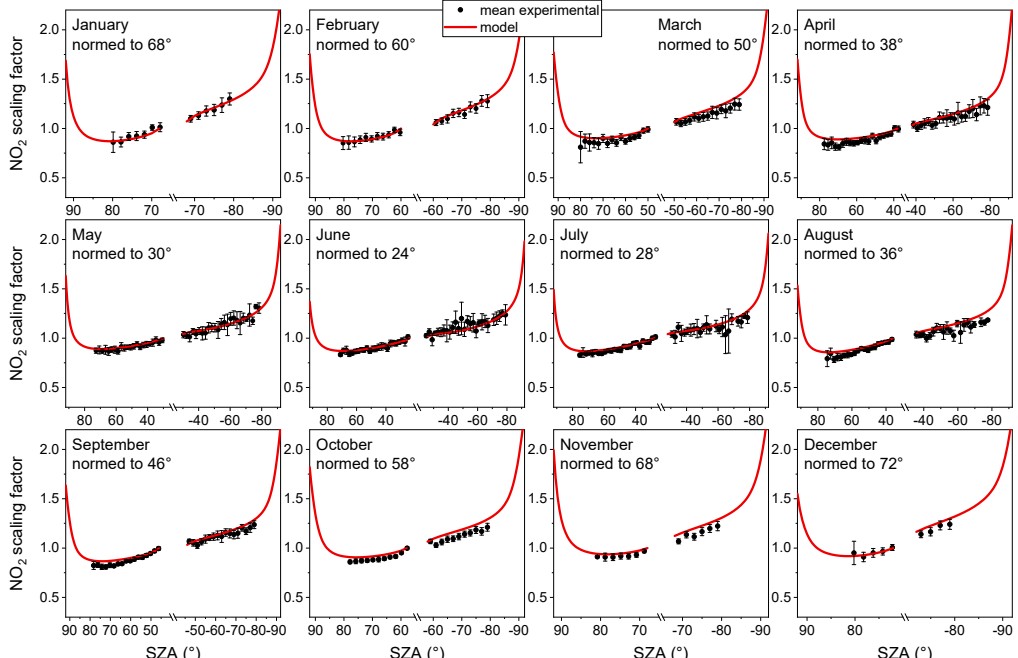

**Figure 5.** Calculated normed $NO_2$ scaling factors $SF_{exp}(NO_2)$ above 16 km altitude measured at Zugspitze (black) and recalculated normed $NO_2$ scaling factors $SF_{sim}(NO_2)$ above 16 km altitude (red line) for every month in dependence of SZA. The experimental values represent the mean value within 2° SZA bins. The error bars represent two times the standard error of the mean ($\pm 2\ \sigma/\sqrt{(n)}$) value. The SZA used for normalization for the respective month for experiment and model is given in each legend.




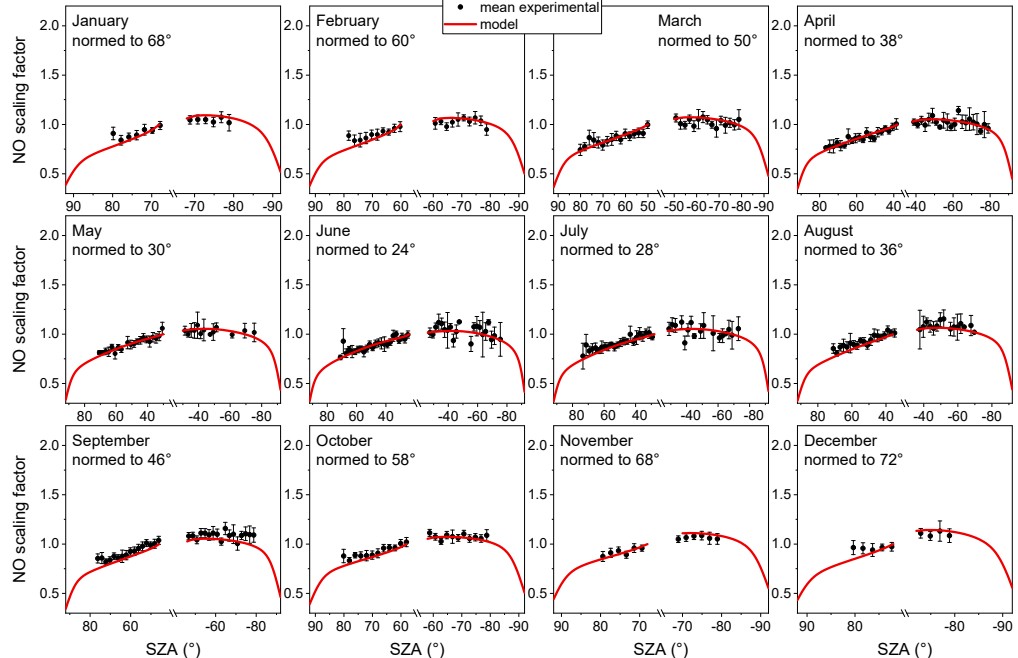

**Figure 6.** Calculated normed NO scaling factors $SF_{exp}$(NO) above 16 km altitude measured at Zugspitze (black) and recalculated normed NO scaling factors $SF_{sim}$(NO) above 16 km altitude (red line) for every month in dependence of SZA. The experimental values represent the mean value within 2° SZA bins. The error bars represent two times the standard error of the mean ($\pm 2 \, \sigma/\sqrt{(n)}$) value. The SZA used for normalization for the respective month for experiment and model is given in each legend.



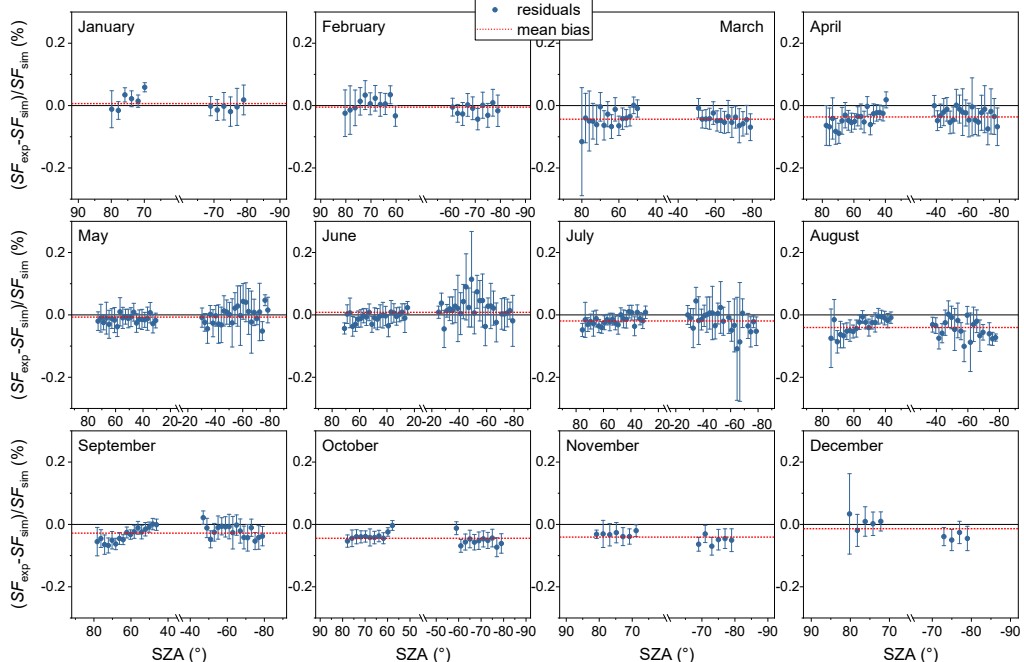

**Figure 7.** Calculated residuals $(SF_{exp}-SF_{sim})/SF_{sim}$ between the experimental normed mean $NO_2$ scaling factors $SF_{exp}$ and the simulated normed $NO_2$ scaling factors $SF_{sim}$ and interpoled to the respective SZA for every month in dependence of SZA. The error bars represent two times the propagated standard error of the mean ($\pm 2\ \sigma/\sqrt{(n)}$) of the experimental value. The mean bias over all SZA is shown in red.

355



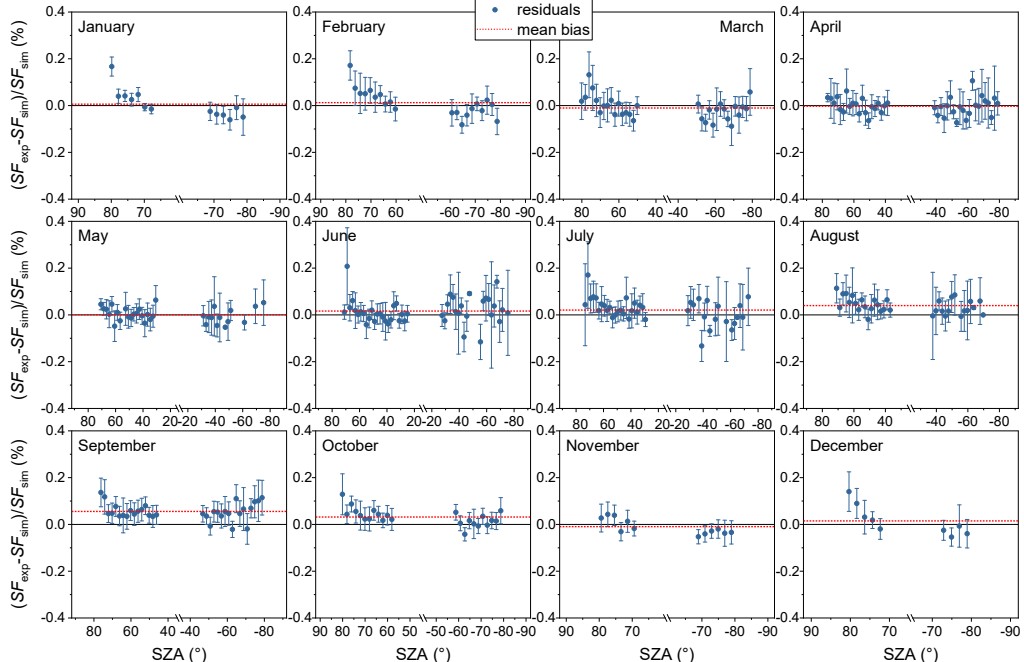

**Figure 8.** Calculated residuals ($[SF_{exp}-SF_{sim}]/SF_{sim}$) between the experimental normed mean NO scaling factors $SF_{exp}$ and the simulated normed NO scaling factors $SF_{sim}$ and interpoled to the respective SZA for every month in dependence of SZA. The error bars represent two times the propagated standard error of the mean ($\pm 2\ \sigma/\sqrt{(n)}$) of the experimental value. The mean bias over all SZA is shown in red.