# Peer review of "Solar FTIR measurements of NOx vertical distributions Part 2"

_EGUsphere, 2023_

## Referee Comment (RC1)

Review of Solar FTIR measurements of NOx vertical distributions: Part 2) Experiment-based scaling factors describing the diurnal increase of stratospheric NO2 and NO

This paper discusses the creation of diurnal scaling factors for NOx based on NO and NO2 partial columns measured at Zugspitze, Germany. Diurnal scaling factors are needed when comparing NOx from instruments that do not measure at the same local solar time. These observational scale factors are useful for validating the model-based scale factors that are typically used.

Some suggestions about word choice

- Title: calling it a 'diurnal increase of NO2 and NO' is not exactly correct. As you say (e.g. abstract line 19-20), NO does not increase throughout the whole day. It might be better to use a word like "change" or "variation".
- Line 47: Define what is meant by "NO2 diurnal increasing rate". I am also not sure about the choice of the word "diurnal" throughout this paper. I think of diurnal as referring to a 24-hour period, rather than only the hours with daylight. Maybe it would be better to say the "daytime NO2 increase"?

Questions & Comments

- Line 33: The phrase "NOx is a product of industry and traffic in the troposphere" is not clear. Are you referring to aircraft emissions? Or emissions at the surface?
- All figures: it would be useful to include local solar time, in addition to SZA, on the x-axis, so that it is easy to see what time of day the SZA corresponds to in each month.
- Choice of normalization SZA:
    o Does the minimum SZA on the 15th of each month correspond to LST=12:00? Typically, one needs to scale observations from multiple instruments to a common LST or SZA, so it is a bit confusing to represent results that are normalized to a different SZA in each month.
    o I understand that a normalization SZA must be chosen for the purposes of this study and comparing with a model. But what would be useful for a user of the data is the ability to scale NO2 measurements at one LST to NO2 measurements at another LST (both LSTs chosen by the user). Thus, it makes more sense to either provide scale factors normalized to every single SZA bin, or to just provide the binned and filtered data as a function of SZA without any normalization so that the normalization SZA can be chosen based on the application.
- What does the blue arrow represent in Figure S1?
- Have you considered how the scale factors change on a time scale smaller than months? In Figure S1 there is more NO2 in the second half of April than in the first half. Is the scale factor significantly different for each half of the month? Would it help with your "boundary value problem" to look at scale factors in e.g. 10 day bins? Dube et al (2022) use model scale factors calculated for each day of the year, while Brohede et al (2007) considered model scale factors

averaged over 2-week periods. It would be interesting to know how important the time binning is- your results suggest that averaging over a full month is not ideal but perhaps averaging over two weeks is adequate, and there is no need to have different scale factors for each day of the year.

Brohede, S. M., Haley, C. S., McLinden, C. A., Sioris, C. E., Murtagh, D. P., Petelina, S. V., ... & Gordley, L. L. (2007). Validation of Odin/OSIRIS stratospheric NO2 profiles. Journal of Geophysical Research: Atmospheres, 112(D7).

Dubé, K., Zawada, D., Bourassa, A., Degenstein, D., Randel, W., Flittner, D., ... & Walker, K. (2022). An improved OSIRIS NO2 profile retrieval in the UTLS and intercomparison with ACE-FTS and SAGE III/ISS. Atmospheric Measurement Techniques Discussions, 2022, 1-22.

- A related question is if the scale factors change over time. Is there enough to data to determine if there is a trend in the scale factors? It would be very interesting if there is a trend as that would suggest some change in NOy chemistry. It is also important for knowing if climatological scale factors are adequate, or if it would be necessary to have scale factors for every year, in addition to every month.
- Discussion of bias between observations and model: Your main explanation for the difference is that the stratospheric partial column is not independent of the tropospheric column. Is there a reason that this would cause the observed structure in the bias (greater bias in the morning)? More discussion should be included about possible issues with the model near sunrise.

Writing edits:

General:

- Some inconsistency with using the roman numeral 'I' and the number '1' to refer to the first paper.
- Inconsistent use of "boundary value problem" and "boundary layer problem"
- Would be good to check the grammar again. I pointed out several things below, but this is not a comprehensive list.

Line 17: Change 'beside' to 'apart from'

Line 19: pluralize 'experiment"

Line 24: remove comma between 'we show' and 'that'

Line 45: "to face" -> "for dealing with"

Line 47: add comma after 'high precision'

Line 55: Should 'received' be 'retrieved'?

Lines 68, 77: "close(d) this lack" -> "close(d) this gap"

Line 71: comma between 'aerosol' and 'radiation'?

Line 92: remove "used"

Line 96: change 'is' to 'has'

Line 105: "not existing" -> "without observations"

Line 121: "in analogy" -> "analogous"

Line 123, 136: "at month 15$^{th}$" –> "on the 15$^{th}$ day of each month,"

Line 128: "follows every month a linear diurnal trend" -> "increases linearly throughout the day in each month"

Line 134: remove comma after 'both'

Line 151: add a comma before 'as', remove word 'before'

Line 175: What is meant by "the whole season"?

Line 182: missing table number

Line 222: remove 'recently' (or add word 'calculated' after 'recently')

---

## Author Comment (AC1)

**Response to Anonymous Referees on acp-2023-1437**

**Solar FTIR measurements of NOx vertical distributions - Part 2: Experiment-based scaling factors describing the daytime variation of stratospheric NO2 and NO**

We thank the Reviewers for their comments and suggestions. Below we provide our answers to their specific comments and the details of the changes made to the revised manuscript.

**Response to Anonymous Referee 1**

*1) As remarked by referee #2, the choice to normalize with a different SZA for each month is inconvenient. It's not a show stopper, but normalizing by a fixed SZA (at Zugspitze there must be one which is covered every single day of the year), would have made more sense to me.*

    We thank Reviewer #1 for this comment.

    We now normalized all data to SZA = 72°. The overall message and the outcome of the manuscript is not influenced by this change. The deviation between experiment and simulation just shift with the time/SZA of normalization. However, with the fixed normalization SZA, the comparability to other data is way easier.

*2) Could you not have avoided the sampling issues (the bias due to only half of the month contributing before or after the 15th in spring/autumn) by normalizing the data for each day to a fixed SZA and then taking the monthly mean (instead of the reverse order as you do now)? In that way the seasonal variation in absolute NOx levels would have been taken out before the monthly averaging.*

    We thank Reviewer #1 for this comment.

    Here, the new data set considers the comment of Reviewer #1. It does not seem to have a big impact, but this way of normalization is more straight forward.

*- Why the cut-off at 16km, which is well above the tropopause at mid latitudes? If justified in Part 1, please briefly repeat the justification here.*

    For a better understanding, we added in line 86-87 of the revised manuscript the following sentences:

    The cut-off point at 16 km was chosen to avoid influences of variabilities near the tropopause and in the boundary layer upon the stratospheric partial column. Details are discussed in part 1.

*- I find the use of negative SZA between noon and sunset somewhat confusing. I think you could have dropped the "minus" in the graphs and just annotated left and right with "AM" and "PM". But ok, no real need to change this.*

    We thank Reviewer #1 for this comment.

Here we decided to be consistent with the paper of Strode et al. (2022) using negative SZA values.

*- The experimental scaling factors are limited to true daytime, excluding the twilight regimes at sunrise and sunset. This is a limitation inherent to the measurement technique, but I think it should be made explicit that the strong and fast photochemistry at sunrise and sunset is outside the scope of these experimental scaling factors.*

We thank Reviewer #1 for this comment.

We added in line 88-89 of the revised manuscript the following sentence:

It is outside the scope of this work to describe with $SF_{exp}$ the strong and fast photochemistry at sunrise and sunset.

*- The poorer comparison to the modelled scaling factors at high SZA: to what extent does your FTIR retrieval take into account the wide range in photochemical regimes along the line-of-sight at these high SZA: high up in the atmosphere, the sun is already well above the horizon, so NO2 loss has been significant already, while lower down the atmosphere is still much darker and NO2 levels still higher. Is that taken into account in the FTIR retrieval, and if so, how? If already discussed in Part 1, please summarize here as well.*

We thank Reviewer #1 for this comment.

With the new normalization method to SZA = 72°, this effect is no more seen for all data, because both data sets are normalized to an earlier time of the day, leading to less deviation of the data in the morning. Only for NO, strong deviations are still seen in the morning.

Therefore, we added in line 201-206 of the revised manuscript the following sentence:

Here, an error source of the experimental data can be the wide range in photochemical regimes along the line-of-sight of the FTIR slant column measurements at high SZA: high up in the atmosphere, the sun is already well above the horizon, so NO production has been significant already, while lower down the atmosphere is still much darker and NO levels still lower. The FTIR retrieval leads to an averaging over these effects because from the solar measurements NO slant columns along the line of sight are retrieved, and these are then converted to vertical column densities using a simple cos(SZA) airmass correction.

*- Also related to the retrieval: Does the stratospheric temperature affect your retrievals (e.g., through the NOx cross sections) and so potentially the observed diurnal cycle? Please spend a few words on this.*

We thank Reviewer #1 for this comment.

We added in line 100-103 of the revised manuscript the following sentence:

As described in part 1, we used daily pressure and temperature profiles from the National Centers for Environmental Prediction (NCEP) interpolated to the measurement time. The temperature dependency of the data cannot be discussed in detail here, but it is very likely that the stratospheric temperature affects the NO$_x$ concentration and therefore also the observed diurnal cycle.

*- a spell check is needed. I counted several already in the abstract.*

Done

*- abstract: mean bias -> mean value*

Done

*- line 123: "month 15th"-> 15th day of the month? Confirmed by a somewhat redundant sentence a couple of lines further on.*

Done

*- I still think "normed" should be "normalized".*

Done

**Response to Anonymous Referee 2**

*Title: calling it a 'diurnal increase of NO2 and NO' is not exactly correct. As you say (e.g. abstract line 19-20), NO does not increase throughout the whole day. It might be better to use a word like "change" or "variation".*

The title of the revised manuscript has been modified accordingly:

Solar FTIR measurements of NOx vertical distributions: Part II) Experiment-based scaling factors describing the daytime variation of stratospheric NOx

*- Line 47: Define what is meant by "NO2 diurnal increasing rate". I am also not sure about the choice of the word "diurnal" throughout this paper. I think of diurnal as referring to a 24-hour period, rather than only the hours with daylight. Maybe it would be better to say the "daytime NO2 increase"?*

We thank Reviewer #2 for this comment.

The word "diurnal" was replaced by "daytime" in the revised manuscript

*- Line 33: The phrase "NOx is a product of industry and traffic in the troposphere" is not clear. Are you referring to aircraft emissions? Or emissions at the surface?*

Text in line 33-34 of the revised manuscript has been modified accordingly:

Additionally, industry and transportation are major sources of tropospheric NOx NOx is a product of industry and traffic in the troposphere (Grewe et al., 2001).

*- All figures: it would be useful to include local solar time, in addition to SZA, on the x-axis, so that it is easy to see what time of day the SZA corresponds to in each month.*

The figures of the revised manuscript have been modified

*- Choice of normalization SZA:*
- *Does the minimum SZA on the 15th of each month correspond to LST=12:00? Typically, one needs to scale observations from multiple instruments to a common LST or SZA, so it is a bit confusing to represent results that are normalized to a different SZA in each month.*

We thank Reviewer #2 for this comment.

We now normalized all data to SZA of 72°. Of course, it results in other and bigger deviations from the model we additionally discussed in the manuscript due to the differing normalization daytime. However, with the fixed normalization SZA, the comparability to other data is way easier.

*- I understand that a normalization SZA must be chosen for the purposes of this study and comparing with a model. But what would be useful for a user of the data is the ability to scale NO2 measurements at one LST to NO2 measurements at another LST (both LSTs chosen by the user). Thus, it makes more sense to either provide scale factors normalized to every single SZA bin, or to just provide the binned and filtered data as a function of SZA without any normalization so that the normalization SZA can be chosen based on the application.*

We thank Reviewer #2 for this comment.

We now provide the binned and filtered data set as a function of SZA with and without normalization in the supplement.

*- What does the blue arrow represent in Figure S1?*

Text in line 145-146 of the revised manuscript has been modified accordingly:

Consequently, the partial column and of course the scaling factor increases artificially (pointed out by the blue arrow in the figure).

*- Have you considered how the scale factors change on a time scale smaller than months? In Figure S1 there is more NO2 in the second half of April than in the first half. Is the scale factor significantly different for each half of the month? Would it help with your "boundary value problem" to look at scale factors in e.g. 10 day bins? Dube et al (2022) use model scale factors calculated for each day of the year, while Brohede et al (2007) considered model scale factors averaged over 2-week periods. It would be interesting to know how important the time binning is- your results suggest that averaging over a full month is not ideal but perhaps averaging over two weeks is adequate, and there is no need to have different scale factors for each day of the year.*

We thank Reviewer #2 for this comment.

Here, we decided to stay at the time-binning of months due to i) the comparability to the simulation-based data and ii) the larger data base which leads to significant trends, which maybe cannot be observed with smaller time bins. However, we included in line 147-149 the following sentence facing this problem:

Another opportunity to face this problem would be the choice of a smaller time binning (e.g. 2 weeks, 10 days). However, this would i) worsen the comparability to the simulation-based scaling factors and ii) reduce the usable data base per time bin.

*- A related question is if the scale factors change over time. Is there enough to data to determine if there is a trend in the scale factors? It would be very interesting if there is a trend as that would suggest some change in NOy chemistry. It is also important for knowing if climatological scale factors are adequate, or if it would be necessary to have scale factors for every year, in addition to every month.*

We thank Reviewer #2 for this comment.

This is a very interesting question. Luckily, the data base is sufficient, to determine monthly variations in diurnal NO2 increase as can been seen in the paper and which are discussed especially in part 1. For the determination of trends over several years., the data base unfortunately is not big enough.

*- Discussion of bias between observations and model: Your main explanation for the difference is that the stratospheric partial column is not independent of the tropospheric column. Is there a reason that this would cause the observed structure in the bias (greater bias in the morning)? More discussion should be included about possible issues with the model near sunrise.*

We thank Reviewer #2 for this comment.

Text in line 211-212 of the revised manuscript has been modified accordingly:

Consequently, the lower stratospheric partial column in the morning is more influenced by the tropospheric partial column than in the evening.

Text in line 188-189 of the revised manuscript has been modified accordingly:

*Furthermore, the model data offer higher uncertainties during twilight which can lead to deviations from experiment (Alvanos and Christoudias, 2019).*

*Writing edits:*

*General:*
*- Some inconsistency with using the roman numeral 'I' and the number '1' to refer to the first paper.*

We thank Reviewer #2 for this comment.

The manuscript has been modified accordingly.

*- Inconsistent use of "boundary value problem" and "boundary layer problem"*

We thank Reviewer #2 for this comment.

The manuscript has been modified accordingly. We used the wording "boundary value" throughout the manuscript.

*- Would be good to check the grammar again. I pointed out several things below, but this is not a comprehensive list.*

Done

*Line 17: Change 'beside' to 'apart from'*
*Line 19: pluralize 'experiment"*
*Line 24: remove comma between 'we show' and 'that'*
*Line 45: "to face" -> "for dealing with"*
*Line 47: add comma after 'high precision'*
*Line 55: Should 'received' be 'retrieved'?*
*Lines 68, 77: "close(d) this lack" -> "close(d) this gap"*
*Line 71: comma between 'aerosol' and 'radiation'?*
*Line 92: remove "used"*
*Line 96: change 'is' to 'has'*
*Line 105: "not existing" -> "without observations"*
*Line 121: "in analogy" -> "analogous"*
*Line 123, 136: "at month 15th*
*" –> "on the 15th day of each month,"*
*Line 128: "follows every month a linear diurnal trend" -> "increases linearly throughout the day in each month"*
*Line 134: remove comma after 'both'*
*Line 151: add a comma before 'as', remove word 'before'*
*Line 175: What is meant by "the whole season"?*
*Line 182: missing table number*
*Line 222: remove 'recently' (or add word 'calculated' after 'recently'*

Done

---

## Author Response (AR2)

**Response to Anonymous Referees on acp-2023-1437**

**Solar FTIR measurements of NOx vertical distributions - Part 2: Experiment-based scaling factors describing the daytime variation of stratospheric NO2 and NO**

We thank the Reviewer #2 for the technical correction. Below we provide our answers to their specific comments and the details of the changes made to the revised manuscript.

**Response to Anonymous Referee 2**

- Is there a reason for choosing to normalize to 72 degrees instead of some other SZA? It would be good to mention this.

    We thank Reviewer #1 for this comment.

    We added in line 129-130 of the revised manuscript the following sentence:

    (which is the only value being present in all monthly data sets)

- line 156: 'originated' should be 'originating'

    Done

- line 203: change 'so NO production has been significant already' to 'so there has already been significant NO production'

    Done

- line 204: change to 'levels are still lower'

    Done

- line 232: still says you normalized to the 15th day of the month, change to 72 degrees

    Done